# Electrochemical Synthesis of Methoxy-*NNO*-azoxy Compounds via N=N Bond Formation Between Ammonium *N*-(methoxy)nitramide and Nitroso Compounds

**DOI:** 10.3390/molecules30244723

**Published:** 2025-12-10

**Authors:** Alexander S. Budnikov, Andrey A. Kulikov, Michael S. Klenov, Nikita E. Leonov, Igor B. Krylov, Alexander O. Terent’ev, Vladimir A. Tartakovsky

**Affiliations:** N. D. Zelinsky Institute of Organic Chemistry, Russian Academy of Sciences, 47 Leninsky Prosp., 119991 Moscow, Russia; alsbudnikov@gmail.com (A.S.B.); aakulikov@ioc.ac.ru (A.A.K.); leonovne@ioc.ac.ru (N.E.L.); terentev@ioc.ac.ru (A.O.T.); tva@ioc.ac.ru (V.A.T.)

**Keywords:** electrochemistry, N–N coupling, azoxy compounds, nitroso compounds

## Abstract

In this study, atom- and step-efficient electrochemical coupling of nitroso compounds with ammonium *N*-(methoxy)nitramide, furnishing methoxy-*NNO*-azoxy compounds, is reported. The developed protocol employs a divided electrochemical cell, proceeds under constant-current electrolysis conditions, and is applicable to aromatic, heterocyclic, and aliphatic nitroso compounds. The versatility of the developed electrochemical coupling method was demonstrated by comparing it with chemical approaches on various substrates.

## 1. Introduction

Organic compounds containing N–N and N–O bonds are widely used as sources of free radicals for selective transformations [1,2,3,4,5,6,7] and polymerization initiators [8], energetic materials [9,10,11], and HNO donors [12,13]. Despite the practical importance of such compounds, synthetic approaches to the construction of N–N and N–O systems are limited in number and application scope. Selective N–N [1,14,15,16,17,18,19,20,21,22,23,24,25] and N–O [26,27,28,29,30,31] coupling is almost unexplored compared to C–C and C–Het coupling processes.

To date, electro-organic synthesis has become a powerful and reliable strategy for the functionalization of organic compounds under green and mild reaction conditions [32,33,34,35]. However, electrochemical N–N and N=N bond formation remains undeveloped. The vast majority of developed approaches focus mainly on intramolecular radical cyclizations [36,37,38,39,40,41,42,43,44,45,46], while intermolecular [14,47,48,49,50,51] N–N coupling remains a poorly studied area.

The *O*^2^-Methylated diazeniumdiolate functional group [–N(O)=N–OMe], also known as the methoxy-*NNO*-azoxy moiety, has been known for over a century since Traube’s pioneering work [52]. Since then, this functional group has attracted considerable attention, starting with the elucidation of its structure [53,54,55], and continuing with the development of novel synthetic routes [56,57,58,59,60,61,62,63], biological evaluation and NO-release profiling [64,65,66], and the design of the novel energetic materials [67,68,69].

The diversity of potential applications of compounds containing the methoxy-*NNO*-azoxy moiety necessitates the development of versatile synthetic methods for the construction of this group. However, to date, only three approaches to the synthesis of methoxy-*NNO*-azoxy compounds have been reported, all of which lack versatility (Figure 1).

The first and the oldest approach is based on the methylation of the salts of nitrosohydroxylamines (Figure 1, eq. 1). Though this method is widely used and allows one to synthesize diverse series of compounds, it has major drawbacks. First, the yield of the desired compound strongly depends on the substrate’s structure—aliphatic derivatives are typically obtained in good yields (>50%), whereas yields of heterocyclic compounds are below 20% [63]. Second, the reaction proceeds with the formation of two isomers, thereby reducing product yield and complicating the purification of the target compound.

The second approach involves the coupling of nitroso compounds with methoxyamine in the presence of an oxidant (Figure 1, eq. 2). The typical oxidants used in this method include lead tetraacetate (Pb(OAc)_4_) [57,60], di(acetoxy)iodobenzene (PhI(OAc)_2_) [60,69], dibromoisocyanuric acid (DBI) [60,69], *N*-bromosuccinimide (NBS) [60], and bromine [60]. Although this method enables the synthesis of a wide range of methoxy-*NNO*-azoxy compounds, their yields typically do not exceed 50% and are accompanied by the formation of several byproducts.

The third approach recently discovered by our research team utilizes the nucleophilic addition–elimination reaction of ammonium *N*-(methoxy)nitramide with 3-amino-4-nitrosofurazan, which furnishes 3-amino-4-(methoxy-*NNO*-azoxy)furazan (Figure 1, eq. 3) [69]. This method has proven to be a convenient route for the construction of the methoxy-*NNO*-azoxy group and produces a promising yield; however, its applicability has not been evaluated with other substrates.

Previously, we have developed an efficient method for the synthesis of (nitro-*NNO*-azoxy)arenes [Ar–N(O)=N–NO_2_] via the electrochemical coupling of nitrosobenzenes with ammonium dinitramide [70]. Consequently, the application of a similar electrochemical approach to the preparation of methoxy-*NNO*-azoxy compounds appeared promising.

Herein, we report a versatile electrochemical approach to synthesizing methoxy-*NNO*-azoxy compounds, employing ammonium *N*-(methoxy)nitramide as the reagent and nitroso compounds as coupling partners (see Figure 1, bottom). The usefulness of this novel method is demonstrated by comparing it with previously described chemical approaches (Figure 1, eqs. 2 and 3) on aliphatic, aromatic, and heterocyclic substrates.

## 2. Results and Discussion

First, the traditional approach to the synthesis of methoxy-*NNO*-azoxy compounds via the oxidative coupling of nitroso compounds with methoxyamine was studied (Figure 1, eq. 2). The reaction conditions were optimized for the model reaction of nitrosobenzene **1a** with methoxyamine in the presence of a common oxidant for such transformations—PhI(OAc)_2_ (Figure 2).

Carrying out the reaction in dichloromethane for 2 h at 23–25 °C with the equimolar amounts of methoxyamine and PhI(OAc)_2_ turned out to be the most optimal, and (methoxy-*NNO*-azoxy)benzene **2a** was obtained in 39% yield. The variation in solvent (MeCN, Et_2_O), temperature (0 °C, 40 °C), oxidant (DBI), and stoichiometric ratio of oxidant/methoxyamine/**1a** (2:1:1, 2:2:1, 4:2:1) either had no effect on the product yield or led to its decrease. This reaction proceeds with the formation of several unidentified byproducts, which substantially complicate the isolation and purification of compound **2a**.

Subsequently, an alternative approach to synthesizing methoxy-*NNO*-azoxy compounds was investigated, involving the reaction of nitroso compound **1a** with ammonium *N*-(methoxy)nitramide **3** [69] (see Figure 1, eq. 3). Optimization of the reaction conditions was carried out by employing nitrosobenzene **1a** as a model substrate (Table 1). Reaction completion was monitored by TLC analysis until full consumption of starting compound **1a**.

Initially, the optimal solvent was identified (Table 1, entries 1–11). The use of a MeCN/MeOH (1/1) mixture afforded compound **2a** in the highest yield (30%, entry 11). Conducting the reaction in DMSO resulted in a slightly lower yield, although with a reduced overall reaction time (entry 4). Next, we tested the possibility of accelerating the reaction via heating (entries 12, 13). However, carrying out the reaction at 50 °C led to a lower yield of the desired product. Although optimization of this method achieved a yield comparable to that of the oxidative coupling method (see Figure 2), the reaction time was significantly increased.

Presumably, the low efficiency of the reaction between **1a** and methoxyamine or **3** is associated with insufficient electrophilicity of **1a**. Thus, we decided to try a fundamentally different approach to intermolecular N=N coupling. Our idea is based on the generation of an electrophilic *N*-centered radical from **3** via anodic oxidation (Figure 1, bottom). Previously, such a strategy was barely explored, except for N=N coupling between nitrosoarenes and ammonium dinitramide [70]. However, employing conditions suitable for ammonium dinitramide [70] for the case of ammonium *N*-(methoxy)nitramide **3** in the synthesis of **2a** yielded the desired product in only 10% (Table 2, entry 1). Variation in solvent and electrode failed to significantly improve the yield of **2a** (Table 2, entries 2–6). The low yield of the product can be attributed to competing cathodic processes.

Therefore, the reaction conditions for the reaction between **1a** and **3** were optimized using a divided electrochemical cell. The influence of the electrode material, amount of electricity, current density, supporting electrolyte, and solvent was evaluated (Table 3).

The solvent was varied first (Table 3, entries 1–9). The best result was obtained using DMSO as a solvent despite its redox-active nature in electrochemistry [71] (entry 9). A comparable result was obtained with a THF/H_2_O system (Table 3, entry 6); however, nitrobenzene **4a** was formed in a higher yield. Employing other solvents (Table 3, entries 1–5, 7, 8) was ineffective due to over-oxidation of nitrosobenzene **1a** to nitrobenzene **4a**. Next, we varied the quantity of **3** (Table 3, entries 10–11). An amount of 1 mmol of **3** per 0.5 mmol of **1a** was found to be the optimal (Table 2, entry 10). The supporting electrolyte was also tested (Table 3, entries 12–16) and the best result was obtained with TBAB (*n*-Bu_4_NBr, Table 3, entry 15). A comparable yield of **2a** was also achieved by employing LiClO_4_ as an electrolyte (Table 3, entry 12); however, lithium reduction on the cathode surface was observed. As expected for a divided cell, the cathode material was found to have a negligible effect on the reaction outcome (Table 3, entries 17–18), and a stainless steel (SS) cathode was chosen for its availability. In contrast, the anode material was crucial (Table 3, entries 20–23). Only 10% of **2a** was obtained with a nickel anode (Table 3, entry 20), while 75%, 70%, and 78% of **2a** was obtained with graphite (Table 3, entry 21), carbon felt (Table 3, entry 22), and glassy carbon (Table 3, entry 23) anodes. Therefore, GC was chosen as an optimal anode material. A decrease in current density (Table 3, entries 24–26) resulted in lower yields compared to entry 23, while an increase led to a high voltage. Investigation of the charge requirement revealed that 2 F mol^−1^ of **1a** was optimal (Table 3, entry 23). Deviating from this optimum, either higher or lower, reduced the yield of **2a** (Table 3, entries 27–29).

With the optimal reaction conditions identified (Table 3, entry 22), the scope for electrochemical synthesis of methoxy-*NNO*-azoxy compounds was tested (Figure 3, Method A). In parallel, the same series of methoxy-*NNO*-azoxy compounds **2a**–**s** were synthesized via oxidative coupling and nucleophilic addition–elimination methods (Figure 3, Methods B and C), enabling a comprehensive comparison and the identification of any systematic patterns.

The developed electrochemical methoxy-*NNO*-azoxylation protocol (Method A) was found to be applicable to a range of electron-deficient (**1b**–**d**,**j**,**k**), electron-rich (**1e**–**g**), and halogen-substituted (**2h**,**i**) nitrosobenzenes, affording the corresponding products in moderate to good yields while maintaining short reaction times (~1 h on a 0.5 mmol scale). The highest yields (51–92%) were obtained for compounds **2b**–**d**,**h**,**i**,**k** prepared from nitrosobenzenes **1b**–**d**,**h**,**i**,**k** containing either an electron-withdrawing nitro or trifluoromethyl group, or a halogen substituent. The decrease in yield of *ortho*-substituted products (**2b**, **2h**–**2j**) could be attributed to the steric hindrance around the nitroso group caused by the –NO_2_, –CF_3_ groups or halogen atoms. In the case of the OMe group, the higher yield of the *meta*-substituted product **2f** (87%) compared to the *ortho*- and *para*-substituted products **2e** (51%) and **2g** (33%) can be explained by the positive mesomeric effect of the electron-donating methoxy group, which leads to a decrease in the oxidation potential of the corresponding nitroso compounds **1e** and **1g** and consequently its side processes of anodic oxidation. At the *meta*-position, the methoxy group does not conjugate with the nitroso group and thus predominantly acts as an electron-withdrawing group via the inductive effect providing the high yield of **2f**, which is in agreement with its slightly positive Hammett constant σ_m_ about +0.11. However, there is no strict correlation between the electronic effects of substituents in *meta*- or *para*-positions and the yields of the corresponding products **2**. The yield is non-monotonously distributed in a series of substituents from the most electron-accepting to the least electron-accepting: *p*-NO_2_ (σ_p_ = +0.78, **2d** yield 80%), *m*-NO_2_ (σ_m_ = +0.71, **2c** yield 92%), *m*-CF_3_ (σ_m_ = +0.43, **2k** yield 60%), and *m*-OMe (σ_m_ = +0.11, **2f** yield 87%). Apparently, there are several other factors determining the yield of target products **2** in electrochemical synthesis in addition to the major factor of nitrosoarenes’ stability in response to anodic oxidation, which may include the rate constants for the interception of the *N*-(methoxy)nitramide-derived radical, the tuning of a substituent electronic effect due to hydrogen bond formation, and side processes associated with excessive electrophilicity.

The reaction of electron-deficient nitrosobenzenes (**1b**–**d**,**j**,**k**) with the salt **3** in acetonitrile-methanol solution in the absence of the electrical current (see Figure 3, Method B) also afforded the corresponding methoxy-*NNO*-azoxy compounds **2b**–**d**,**j**,**k**; however, the reaction time was significantly longer (3–4 days) and the yields were lower (5–43%) compared to Method A. Unlike their electron-deficient counterparts, electron-rich nitrosobenzenes (**1e**–**g**) showed no reaction with **3** in the absence of an electrical current.

It should be noted that oxidative coupling of nitrosobenzenes **1a**–**k** with methoxyamine in the presence of PhI(OAc)_2_ in dichloromethane (see Figure 3, Method C) afforded the entire scope of substituted benzenes **2a**–**k**; however, the yields in each case were lower (14–43%) than those obtained using Method A.

The discovered electrochemical reaction (Method A) also demonstrated tolerance toward aliphatic nitroso compounds **1l**–**p**, which contain a nitro group in the geminal position, affording the corresponding products **2l**–**p,** albeit in low yields (21–39%). Notably, 2,2-dimethyl-5-nitro-5-nitroso-1,3-dioxane **1p** was successfully used for the synthesis of **2p** in 35% yield without cleavage of the acetonide-protecting group. Unlike for aromatic nitroso compounds, the reaction of aliphatic nitroso compounds without any current (Method B) resulted in higher yields for some products compared to Method A. Specifically, the yield of **2l** and **2p** increased from 26% to 35% and from 35% to 42%, respectively. However, the reaction time was also increased to two days. Application of Method C also offered an increase in the yields of **2l** and **2p**, although to a slightly lesser extent than Method B (30% and 38%, respectively).

Furthermore, the electrochemical methoxy-*NNO*-azoxylation protocol was successfully extended to heteroaromatic systems, namely 2-nitrosopyridine (**1q**), 2-methyl-3-nitroso-5-nitro-*2H*-triazole (**1r**), and 2-methyl-5-nitroso-*2H*-tetrazole (**1s**), yielding the corresponding products **2q**–**s**. The yield of the pyridine derivative **1q** was 61%, which is comparable to that of benzene derivatives **2a**–**f**, although (methoxy-*NNO*-azoxy)azoles **2r** and **2s** were obtained in low yields. In the case of the tetrazole derivative **1s**, in addition to the target methoxy-*NNO*-azoxy product **2s**, the known 5,5′-azoxy-bis-2,2′-methyltetrazole (**5**) [72] was also isolated in 40% yield. Application of Method B enabled an increase in the yield of **2s** from 28% to 51%, probably due to the mitigation of the competing process of formation of azoxytetrazole **5**. Method C exhibited the poorest results among the studied methods, as the yields of azoles **2r** and **2s** did not exceed 5%.

To clarify the mechanism of the electrochemical methoxy-*NNO*-azoxylation reaction, cyclic voltammetry (CV) studies were performed in DMSO solution using a glassy carbon working electrode, with tetrabutylammonium tetrafluoroborate as the supporting electrolyte (Figure 1).

Figure 1 demonstrates that ammonium *N*-(methoxy)nitramide **3** is the most readily oxidizable component in the reaction mixture, exhibiting a low oxidation potential of approximately 424 mV alone (Figure 1 (b)) and 379 mV in a reaction mixture with **1a** and *n*-Bu_4_NBr (Figure 1 (c)). Oxidation of the bromide anion occurs at approximately 815 mV (Figure 1 (Background, grey)); however, upon addition of nitrosobenzene, it appears to be co-oxidized with 1a (715 mV, Figure 2 (a)).

To elucidate the oxidation potentials of reagents and products in the absence of *n*-Bu_4_NBr, cyclic voltammetry (CV) measurements were additionally performed using only *n*-Bu_4_NBF_4_ as the supporting electrolyte (Figure 2).

Cyclic voltammetry (CV) data indicate that ammonium *N*-(methoxy)nitramide **3** remains the most readily oxidizable component in the reaction mixture, with a low oxidation potential of approximately 330 mV (Figure 2 (b) and (d)). The integration of curve b corresponding to the anodic oxidation of *N*-(methoxy)nitramide **3** gave almost the same value as that obtained for the integration of the oxidation curve of FeCp_2_ of the same concentration. This result indicates that the observed oxidation peak of **3** corresponds to a one-electron oxidation, leading to the formation of a free radical from the N-(methoxy)nitramide anion. In contrast, nitrosobenzene **1a** (Figure 1 (a)) and the reaction product **2a** (Figure 1 (c)) show no noticeable oxidation peaks.

To gain further insight into the reaction mechanism, additional control experiments were conducted (Figure 4).

The obtained results (Table 1) indicate that in the case of an aromatic nitroso compounds without electron-withdrawing groups, the reaction proceeds very slowly, suggesting a distinct mechanism for the electrochemical transformation. To elucidate the requirement for a divided electrochemical cell, the model reaction was performed in an undivided cell under controlled-potential electrolysis (CPE) conditions (Figure 4). The yield of **2a** did not exceed 19%. This result demonstrates that the process occurs at the anode surface, confirming that a separated cathode compartment is essential for high reaction efficiency.

On the basis of CV studies, control experiments, and our previous work concerning the electrochemical behavior of nitrosobenzene and ammonium dinitramide [70], the following mechanism for the formation of methoxy-*NNO*-azoxy compounds 2 from nitroso compounds 1 was proposed (Figure 5).

The reaction is initiated by the anodic oxidation of the *N*-(methoxy)nitramide anion to generate an *N*-centered radical **A**. This radical **A** then reacts with the nitroso compound **1** to form *N*-oxyl radical **C**, which subsequently eliminates a molecule of NO_2_ to yield the final product **2**. Alternatively, compound **2** can be formed via a competing pathway involving α-elimination of NO_2_ from radical **A** to generate methoxy-nitrene **D**, which subsequently reacts with nitroso compound **1**.

For the reaction conducted without electrical current, the proposed mechanism involves nucleophilic addition of the *N*-(methoxy)nitramide anion to the nitroso group, forming intermediate **C′**. Subsequent elimination of the nitrite anion yields the methoxy-*NNO*-azoxy group.

## 3. Materials and Methods

In all experiments, RT stands for 22–25 °C. ^1^H, ^13^C, ^14^N, and ^15^N NMR spectra were recorded with Bruker DRX-500 (500.1, 125.8, 36.1, 50.7 MHz, respectively) and Bruker AV600 (600.1, 150.9, 43.4, 60.8 MHz, respectively) spectrometers (Bruker BioSpin GmbH, Rheinstetten, Germany). Chemical shifts are reported in delta (*δ*) units, parts per million (ppm) downfield from internal TMS (^1^H, ^13^C) or external CH_3_NO_2_ (^14^N, ^15^N negative values of δ_N_ correspond to upfield shifts). The IR spectra were recorded with a Bruker ALPHA-T spectrometer (Bruker Corporation, Billerica, MA, USA) in the range 400–4000 cm^−1^ (resolution 2 cm^−1^) as pellets with KBr or as a thin layer. High-resolution ESI mass spectra (HRMS) were recorded with a Bruker micrOTOF II instrument (Bruker Corporation). Silica gel 60 Merck (Merck, Darmstadt, Germany) (15–40 μm) was used for preparative column and thin-layer chromatography. Silica gel “Silpearl UV 254” was used for preparative column and thin-layer chromatography. Analytical thin-layer chromatography (TLC) was carried out on Merck silica gel 60 F254 (Merck) and “Silufol” TLC silica gel UV-254 (Kavalier, Votice, Czech Republic) aluminum sheets. All reagents were purchased from Acros (Waltham, MA, USA) and Sigma-Aldrich (St. Louis, MO, USA). Solvents were purified before use, according to standard procedures. All other reagents were used without further purification. All electrodes, with the exception of the carbon felt (CF), were flat, polished plates for which the geometrical surface area was equal to the electrochemically active surface area. Commercial carbon felt (PANCF3200300, derived from polyacrylonitrile, ≥98% carbon content, 3 mm thickness) was used as is.

### 3.1. The General Procedure for the Optimization of the Reaction Conditions for the Synthesis of 1-(Methoxy-NNO-azoxy)benzene (***2a***) from 1-Nitrosobenzene (***1a***) via Oxidative Coupling (Experimental Details for Figure 2)

To a stirred suspension of **1a** (59 mg, 0.50 mmol) and PhI(OAc)_2_ (161–644 mg, 0.50–2.00 mmol) or dimbromoisocyanuric acid (DBI) (144–574 mg, 0.50–2.00 mmol) in dry solvent (2 mL, see Appendix A) at 0 °C under an argon atmosphere, a solution of MeONH_2_ (24–47 mg, 0.50–1.00 mmol) in dry CH_2_Cl_2_ (1 mL) was added dropwise. Then, the reaction mixture was vigorously stirred at 0–40 °C (see Appendix A) for 2 h. In the case of PhI(OAc)_2_, the reaction mixture was concentrated under reduced pressure. When DBI was used, the formed precipitate was filtered off, washed with CH_2_Cl_2_ (3 × 2 mL) and then combined filtrates were concentrated under reduced pressure. The yields of **2a** were determined with the use of ^1^H NMR spectroscopy using 1,1,2,2-tetrachloroethane as an internal standard (see Appendix A).

### 3.2. The General Procedure for the Optimization of the Reaction Conditions for the Synthesis of 1-(Methoxy-NNO-azoxy)benzene (***2a***) from 1-Nitrosobenzene (***1a***) and ***3*** Without Electricity (Experimental Details for Table 1)

To a stirred solution of **1a** (59 mg, 0.50 mmol) in 2 mL DMSO, MeCN, or MeCN/MeOH at 25 °C, ammonium *N*-(methoxy)nitramide (**3**) (55 mg, 0.50 mmol) was added. Then, the reaction mixture was vigorously stirred at 25–50 °C for 3h–9 days. The precipitate was then filtered off and washed with MeCN (2 × 2 mL). The combined filtrates were concentrated under reduced pressure. The yields of **2a** were determined with the use of ^1^H NMR spectroscopy using 1,1,2,2-tetrachloroethane as an internal standard.

### 3.3. The General Procedure for the Screening of the Reaction Conditions in an Undivided Electrochemical Cell for the Synthesis of (Methoxy-NNO-azoxy)benzene ***2a*** from Nitrosobenzene ***1a*** (Experimental Details for Table 2)

An undivided 10 mL electrochemical cell was equipped with a platinum plate, carbon felt, or glassy carbon anode (30 × 15 mm), and a platinum wire (d = 1 mm, l = 113 mm, n_coils_ = 9) or stainless steel (SS) cathode, connected to a DC-regulated power supply. The electrodes were fully immersed, providing a total working surface area (S) of 4.5 cm^2^. A solution of nitrosobenzene **1a** (0.5 mmol, 54 mg), ammonium *N*-(methoxy)nitramide **3** (1–2 mmol, 109–218 mg), and a supporting electrolyte (0–0.5 mmol, 0–164 mg) in 10 mL of solvent (MeCN, MeCN/H_2_O, DMF, MeOH, CH_2_Cl_2_/H_2_O, or DMSO) was subjected to constant-current electrolysis at 60 mA and 23–25 °C with magnetic stirring. After passing a charge of 2 F·mol^−1^ (27 min), the electrodes were washed with CH_2_Cl_2_ (3 × 20 mL). The combined organic phase was washed with H_2_O (20 mL) and brine (20 mL), dried over Na_2_SO_4_, and the solvent was removed in vacuo. The yields of **2a** were determined with the use of ^1^H NMR spectroscopy using 1,1,2,2-tetrachloroethane as an internal standard.

### 3.4. The General Procedure for the Optimization of the Reaction Conditions in a Divided Electrochemical Cell for the Synthesis of (Methoxy-NNO-azoxy)benzene ***2a*** from Nitrosobenzene ***1a*** and ***3*** (Experimental Details for Table 3)

A divided *H*-type electrochemical cell (volume of each compartment ~15 mL, divided with Celgard^®^ 2400 membrane) was equipped with a platinum, nickel, graphite, carbon felt, and glassy carbon plate anode (30 × 15 mm^2^) and a platinum, stainless steel, nickel, and glassy carbon plate cathode (30 × 15 mm^2^), and connected to a DC-regulated power supply. A solution of nitrosobenzene **1a** (0.5 mmol, 54 mg), ammonium *N*-(methoxy)nitramide **3** (0.5–1.5 mmol, 54–163 mg), and a supporting electrolyte (1 mmol, 104–369 mg) in solvent (12 mL) was placed in the anodic compartment of the divided electrochemical cell. The cathodic compartment was filled with a solution of the supporting electrolyte (1 mmol, 104–369 mg) in the same solvent (12 mL). Electrolysis was carried out under a constant current (*I* = 10–30 mA, 1–3 F per mole **1a**) at 23–25 °C under magnetic stirring. After passing 1–3 F∙mol^−1^ of electricity (reaction time 27–160 min), the electrodes were washed with CH_2_Cl_2_ (3 × 20 mL). The combined organic phase was washed with H_2_O (2 × 20 mL), dried over Na_2_SO_4_, and the solvent was removed in vacuo. The yields of **2a** were determined with the use of ^1^H NMR spectroscopy using 1,1,2,2-tetrachloroethane as an internal standard. In run 22, **2a** was isolated using column chromatography on silica gel (*R*_f_ = 0.43, petroleum ether/EtOAc, 3:1) to afford the desired product (53 mg, 70%) as a pale-yellow crystals. mp: 40–41 °C. The synthesized compound **2a** was identical (^1^H, ^13^C, and ^14^N NMR, TLC) to the compound prepared according to the reported procedure [64].

### 3.5. Typical Procedure for Synthesis of (Methoxy-NNO-azoxy)compounds ***2a***–***2s*** (Experimental Details for Figure 3)

**Method A.** Electrolysis was conducted in a divided H-cell (15 mL per compartment) equipped with a Celgard^®^ 2400 membrane, using a glassy carbon anode (30 × 15 mm^2^) and stainless steel cathode (30 × 15 mm^2^) connected to a DC power supply. A solution of nitroso compound **1** (0.5 mmol, 54–135 mg), **3** (1 mmol, 109 mg), and *n*-Bu_4_NBF_4_ (1 mmol, 329 mg) in DMSO (12 mL) was placed in the anodic compartment of the divided electrochemical cell. The cathodic compartment was filled with a solution of *n*-Bu_4_NBF_4_ (1 mmol, 329 mg) in DMSO (12 mL). Electrolysis was carried out under a constant current (*I* = 30 mA, 2 F per mole **1**) at 23–25 °C under magnetic stirring. After passing 2 F∙mol^−1^ of electricity (reaction time 54 min), the electrodes were washed with CH_2_Cl_2_ (3 × 20 mL). The combined organic phase was washed with H_2_O (2 × 20 mL), dried over Na_2_SO_4_, and the solvent was evaporated under reduced pressure. The synthesized products **2a**–**2s** were purified using column chromatography on silica gel (petroleum ether/EtOAc, from 5:1 to 1:1). In the case of compound **1s,** 5,5′-azoxy-bis-2,2′-methyltetrazole (**5**) (21 mg, 40%) was also isolated. This product was identical (TLC, ^1^H and ^13^C NMR, HRMS) to the compound prepared according to the reported procedure [72].

**Method B.** To a stirred solution of **1** (0.50 mmol) in 2 mL of a MeCN/MeOH (1/1) mixture at 25 °C, **3** (55 mg, 0.50 mmol) was added. Then, the reaction mixture was vigorously stirred at this temperature for the given time (see Appendix A). The precipitate was then filtered off and washed with MeCN (2 × 2 mL). The combined filtrates were concentrated under reduced pressure. Products **2a**–**2s** were purified using column chromatography on silica gel.

**Method C.** To a stirred suspension of **1** (0.50 mmol) and PhI(OAc)_2_ (161 mg, 0.50 mmol) in dry CH_2_Cl_2_ (2 mL) at 0 °C under an argon atmosphere, a solution of MeONH_2_ (24 mg, 0.50 mmol) in dry CH_2_Cl_2_ (1 mL) was added dropwise. Then, the reaction mixture was vigorously stirred at 25 °C for 2 h. After reaction completion, the mixture was concentrated under reduced pressure. Products **2a**–**2s** were isolated using column chromatography on silica gel.

### 3.6. Reaction Under Controlled-Potential Electrolysis (Experimental Details for Figure 4)

Controlled-potential electrolysis was performed in an undivided 20 mL electrochemical cell using a glassy carbon anode (30 × 15 mm^2^, S = 4.5 cm^2^), a stainless steel cathode (30 × 15 mm^2^), and a Ag/AgNO_3_ reference electrode. A solution of nitrosobenzene **1a** (0.5 mmol, 54 mg) and **3** (1.0 mmol, 109 mg) in DMSO (12 mL) was electrolyzed at 370 mV (*vs.* Ag/AgNO_3_) at 23–25 °C. Upon the passing of 2.0 F·mol^−1^ of electricity, the electrodes were washed with CH_2_Cl_2_ (3 × 20 mL). The combined organic extract was washed with water (2 × 20 mL), dried (Na_2_SO_4_), and concentrated in vacuo. The yield of **2a** was determined according to ^1^H NMR spectroscopy using 1,1,2,2-tetrachloroethane as an internal standard.

### 3.7. Cyclic Voltammetry Studies

Cyclic voltammetry (CV) was implemented on a PS-30 computer-assisted potentiostat-galvanostat manufactured by “SmartStat” (Chernogolovka, Russia) with a scan rate of 100 mV∙s^−1^. Cyclic voltammetry (CV) experiments were conducted in a 10 mL water-jacketed, five-necked conical glass cell using a standard three-electrode configuration. A typical measurement utilized a 5 mL solution under thermostatic control at 21.0 ± 0.5 °C. The working electrode was a glassy carbon disk (*d* = 3 mm), the counter electrode was a platinum wire, and the reference electrode was Ag/AgNO_3_ (0.1 M in 0.1 M n-Bu_4_NBF_4_/MeCN), connected to the solution via a porous glass diaphragm. All solutions were de-aerated through argon purging prior to measurement, and the experiments were performed under an argon atmosphere. The working electrode was polished before recording each CV curve.

## 4. Conclusions

In conclusion, we have developed a robust and efficient electrochemical method for the synthesis of methoxy-*NNO*-azoxy compounds via the coupling of nitroso compounds with ammonium *N*-(methoxy)nitramide. The disclosed transformation proceeds in a divided electrochemical cell under constant-current electrolysis conditions. The developed protocol is applicable to a broad range of nitroso compounds, including aromatic, heterocyclic, and aliphatic derivatives. Compared to chemical approaches that rely on the use of oxidants or an addition–elimination reaction between nitrosocompounds and *N*-(methoxy)nitramide anion, the developed electrochemical method proceeds rapidly and is applicable to both electron-deficient and electron-rich nitroso compounds.

## Data Availability

The data are contained within the article or Appendix A.

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
