# Peer review of "Electrochemical Synthesis of Methoxy-NNO-azoxy Compounds via N=N Bond Formation Between Ammonium N-(methoxy)nitramide and Nitroso Compounds"

_molecules, 2025, doi:10.3390/molecules30244723_

Round 1
Reviewer 1 Report
Comments and Suggestions for Authors
This article reports on the electrochemical coupling of nitroso compounds with N-(methoxy)nitramide. The authors have developed a green and highly efficient electrochemical synthesis approach that is applicable to the coupling reactions of aromatic, heterocyclic, and aliphatic nitro compounds. There are some issues that should be addressed before it is considered for publication:
1 In the present work of Scheme 1, a new radical intermediate was proposed conceptually. However, no relevant representations are given in the following text to determine the existence of this radical intermediate, such as radical trapping experiments or electron paramagnetic resonance (EPR), which prevents confirmation of radical involvement in the proposed mechanism.
3 In Scheme 3, the substrate scope is insufficient to demonstrate the generality of the methodology; substituents such as halogens, trifluoromethyl, and cyano groups should be included.
4 In Table 2, the screening of reaction conditions for the undivided cell setup is inadequate. Additional studies on electrolytes, electrode combinations, and stoichiometric ratios of reagents are necessary.
5 During the reaction process in Table 3, the byproduct nitrobenzene 4a was observed. However, no relevant structural characterization data were provided in the subsequent text to confirm its presence, such as LC-MS or high-resolution mass spectrometry. It is recommended to supplement the necessary analytical evidence.
6 In scheme 4, since both the optimized electrochemical conditions and substrate scope evaluations were performed in a divided cell, the subsequent controlled potential electrolysis (CPE) and constant current electrolysis (CCE) experiments should also be conducted in a divided cell rather than in the undivided cell system, which has not been thoroughly investigated.
7 The labeling of cyclic voltammetry curves (Figures 1. and 2.) should be standardized for clarity.
Author Response
Referee: 1
Reviewer’s comment: This article reports on the electrochemical coupling of nitroso compounds with N-(methoxy)nitramide. The authors have developed a green and highly efficient electrochemical synthesis approach that is applicable to the coupling reactions of aromatic, heterocyclic, and aliphatic nitro compounds.
Answer: Thank you for reading and positive assessment of our manuscript.
Reviewer’s comment: There are some issues that should be addressed before it is considered for publication:
- In the present work of Scheme 1, a new radical intermediate was proposed conceptually. However, no relevant representations are given in the following text to determine the existence of this radical intermediate, such as radical trapping experiments or electron paramagnetic resonance (EPR), which prevents confirmation of radical involvement in the proposed mechanism.
Answer: Thank you for this important question. We thought a lot about this fundamental complicated question and conducted additional experiments, our considerations are set out below.
First, standard radical trapping experiments are not suitable in our case, as typical free radical scavengers (such as TEMPO and BHT) easily undergo anodic oxidation. Furthermore, it is unclear whether TEMPO is capable of scavenging an N-centered radical, as the resulting adduct would contain an extremely weak N–O bond. C=C bond containing radical scavengers are also not suitable for mechanistic studies here, as electron rich radical acceptors (such as styrene, 1,1-diphenylethylene) can undergo anodic oxidation to cation-radicals followed by nucleophile addition (for example, see Chem. Commun., 2025, 61, 4265 https://doi.org/10.1039/D4CC06306F; Nat. Commun. 10, 4953 (2019). https://doi.org/10.1038/s41467-019-13024-5, Nat. Commun. 9, 3551 (2018). https://doi.org/10.1038/s41467-018-06020-8) producing “quenching products” without the formation of free radicals from the corresponding nucleophiles. Electron-deficient alkenes are relatively stable to anodic oxidation, but can produce “quenching products” as a result of aza-Michael reaction. So, for both electron-rich and electron-deficient alkenes, the results can not be reliable for conclusions about the radical mechanism under oxidative conditions with involvement of nucleophilic precursor (such as N-(methoxy)nitramide) for supposed free radicals.
Second, nitroso compounds themselves are well-known free radical scavengers (e.g., see Chem. Commun., 2020, 56, 13719–13730). Therefore, the successful formation of the target products in the present paper provides indirect evidence for a radical mechanism. Especially revealing fact is the superior yield for electrochemical method A compared to method B (Scheme 3) for the case of nitrosoarenes with low electrophilicity. This shows that method A involves a fundamentally different reaction pathway from method B (which is based on nucleophilic attack of N-(methoxy)nitramide at nitroso group). Moreover, CV results (Figures 1,2) indicate that it is the N-(methoxy)nitramide that undergoes anodic oxidation, not nitrosobenzene. Moreover, easily oxidizable electron-rich nitrosoarenes, such as o- and p-methoxy derivatives gave relatively low yields of target products (Scheme 3, products 2e, 2g) due to side process of oxidation with the nitroarene formation. So, the alternative mechanism involving the formation of nitrosoarene cation-radical and nucleophilic attack of N-(methoxy)nitramide to this cation-radical is extremely unlikely.
Third, integration of the CV curves for ferrocene and ammonium N-(methoxy)nitramide 3 of identical concentrations (0.01 M) under identical conditions of potential sweep showed that the oxidation peak of 3 corresponds to a one-electron process, which also supports the formation of an N-centered radical.
Corresponding comment was added to the manuscript: “The integration of curve b corresponding to the anodic oxidation of N-(methoxy)nitramide 3 gave almost the same value as one obtained for the integration of oxidation curve of FeCp2 of the same concentration. This result indicates that observed oxidation peak of 3 corresponds to the one electron oxidation with the formation of a free radical from N-(methoxy)nitramide anion.”
Forth, direct observation of discussed free radicals by EPR is not possible due to their transient nature even for the case of their generation inside EPR cavity in a specialized electrochemical EPR tube. So, according to the Reviewer’s advice we have performed spin trapping experiment with PBN (N-tert-Butyl-α-phenylnitrone) trap. Upon electrolysis of ammonium N-(methoxy)nitramide 3 in DMSO spin adduct was detected (blue line – experimental EPR spectrum, yellow line - simulation):
The observed signal was successfully simulated with following parameters: g0 = 2.0058, aN = 1.39 mT, aH = 0.23 mT. This result confirms the formation of free-radical species, but it was not included into the manuscript, because it is not possible currently to unambiguously assign this spectrum to a certain spin-adduct structure. Hyperfine splitting constants are solvent-sensitive, and there are not enough literature data for comparison. For example, observed signal parameters are quite close to parameters of methoxy radical spin adduct to PBN in CD3CN: aN = 1.38 mT, aH = 0.23 mT [J. Am. Chem. Soc. 2023, 145, 1, 359–376, https://doi.org/10.1021/jacs.2c10126] but many other spin adducts can display similar parameters [Free Radical Biology and Medicine 1987, 3, 259–303, https://doi.org/10.1016/S0891-5849(87)80033-3].
In conclusion, the formation of target products via the formation of radical A (Scheme 5) is in agreement with CV data and reactivity of various nitrosoarenes and other reaction pathways are much less plausible. However, as Reviewer noted, this new radical intermediate was proposed conceptually as an idea led to the present electrochemical method discovery before all these experimental results were obtained. Our study did not claim to prove the formation of this radical.
Reviewer’s comment: 2. In Scheme 3, the substrate scope is insufficient to demonstrate the generality of the methodology; substituents such as halogens, trifluoromethyl, and cyano groups should be included.
Answer: We thank the Reviewer for this proposal. Corresponding products 2h–k with bromine, chlorine and CF3-substitutients were synthesized and included in Scheme 3. Corresponding discussion of Scheme 3 was also revised.
Old version: “The developed electrochemical methoxy-NNO-azoxylation protocol (Method A) was found to be applicable to both electron-deficient (1b–d) and electron-rich (1e–g) nitrosobenzenes, affording the corresponding products in moderate to good yields, while maintaining short reaction times (⁓ 1 h on a 0.5 mmol scale). The highest yields (53–92%) were obtained for compounds 2b–d prepared from nitrosobenzenes 1b–d containing an electron-withdrawing nitro group. Substitution with an electron-donating methoxy group furnished desired products 2e–g in generally moderate yields (33–87%). The higher yield of meta-substituted product 2f (87%) compared ortho- and para-substituted products 2e (51%) and 2g (33%) can be attributed to negative influence of electron-donating +M effect of methoxy group on the target product yield. Apparently, side processes of anodic oxidation of the nitroso group are favored by the conjugation between it and the electron-donating methoxy group.”
Revised version: “The developed electrochemical methoxy-NNO-azoxylation protocol (Method A) was found to be applicable to a range of electron-deficient (1b–d,j,k), electron-rich (1e–g), and halogen-substituted (2h,i) nitrosobenzenes, affording the corresponding products in moderate to good yields, while maintaining short reaction times (⁓ 1 h on a 0.5 mmol scale). The highest yields (51–92%) were obtained for compounds 2b–d,h,i,k prepared from nitrosobenzenes 1b–d,h,i,k containing either an electron-withdrawing nitro or trifluoromethyl group, or a halogen substituent. The decrease in yield of ortho-substituted products (2b, 2h–2j) could be attributed to the steric hindrance around the nitroso group caused by the –NO2, –CF3 groups or halogen atoms. Substitution with an electron-donating methoxy group furnished the desired products 2e–g in generally moderate yields (33–87%). The higher yield of the meta-substituted product 2f (87%) compared to the ortho- and para-substituted products 2e (51%) and 2g (33%) can be explained by the positive mesomeric effect of the electron-donating methoxy group, which leads to a decrease in the oxidation potential of the corresponding nitroso compound and consequently its side processes of anodic oxidation. At the meta-position, the methoxy group does not conjugated with the nitroso group and thus predominantly acts as an electron-withdrawing group via the inductive effect, which is in agreement with its slightly positive Hammett constant σm about +0.1.”
Reviewer’s comment: 4. In Table 2, the screening of reaction conditions for the undivided cell setup is inadequate. Additional studies on electrolytes, electrode combinations, and stoichiometric ratios of reagents are necessary.
Answer: Thank you for this comment. The relatively small number of experiments in Table 2 can be attributed to several factors. First, the low solubility of ammonium N-(methoxy)nitramide 3 limited the screening of a broad range of solvents commonly used in electrochemistry. Second, the choice of conditions was based upon our previous experience in electrochemical N=N coupling with ammonium dinitramide (Molecules 2024, 29(23), 5563). This included the use of a platinum cathode, chosen for its low hydrogen evolution overpotential to minimize competing reduction processes, and a platinum anode, which had proven effective for oxidation in a related reaction. Furthermore, similar to our previous study, we investigated the use of ammonium N-(methoxy)nitramide 3 as both a reagent and supporting electrolyte, using a 4-fold excess (Table 2, entries 3 and 4). However, these attempts were unsuccessful, resulting in the formation of product 2a in yields not exceeding 20%. We attributed this low yield to competing cathodic processes, which prompted the decision to use a divided electrochemical cell. Consequently, the experiment from Scheme 4 (a), conducted under optimized (Table 3) CCE conditions but in an undivided cell was added to Table 2. This result proved that isolation of cathode from the reaction medium (divided cell) is necessary.
Reviewer’s comment: 5. During the reaction process in Table 3, the byproduct nitrobenzene 4a was observed. However, no relevant structural characterization data were provided in the subsequent text to confirm its presence, such as LC-MS or high-resolution mass spectrometry. It is recommended to supplement the necessary analytical evidence.
Answer: Thank you for this note, corresponding results were added. The formation of nitrosobenze 4a was unambiguously observed by 1H- and 14N NMR spectroscopy. Furthermore, in Table 3 entry 23 nitrobenzene was isolated from crude reaction mixture with 14% yield. NMR spectra of 4a were added to SI (S124–S126).
Reviewer’s comment: 6. In scheme 4, since both the optimized electrochemical conditions and substrate scope evaluations were performed in a divided cell, the subsequent controlled potential electrolysis (CPE) and constant current electrolysis (CCE) experiments should also be conducted in a divided cell rather than in the undivided cell system, which has not been thoroughly investigated.
Answer: Thank you for this note. Constant current electrolysis in a divided electrochemical cell was performed during the optimization of reaction conditions and evaluation of the reaction scope (see Table 3 and Scheme 3). The experiment shown in Scheme 4 was conducted to assess the viability of the optimized reaction conditions from Table 3 in an undivided electrochemical cell. This experiment has been relocated from Scheme 4 to Table 2.
Unfortunately, controlled-potential electrolysis in a divided electrochemical cell could not be performed due to high resistance of the divided cell and, consequently, excessively high operating voltages (35 to 40 V with the DC-regulated power supply used in this study), which exceeded the capabilities of the potentiostat used.
Performing this experiment in an undivided cell is nevertheless important because it underscores the complexity of the reaction system. Even when applying the precise oxidation potential of ammonium N-(methoxy)nitramide 3, as derived from cyclic voltammetry (CV) studies (Figures 1 and 2), the yield of product 2a did not exceed 16%. This result indicates the significant influence of cathodic processes and underscores the need to separate the cathodic and anodic compartments.
Reviewer’s comment: 7. The labeling of cyclic voltammetry curves (Figures 1. and 2.) should be standardized for clarity.
Answer: We thank the Reviewer for this essential note. Structures of studied compounds, their oxidation potentials and improved labeling were added in Figures 1 and 2.
Old versions:
(Figure 1)
(Figure 2)
Revised versions:
(Figure 1)
(Figure 2)

Reviewer 2 Report
Comments and Suggestions for Authors
The manuscript describes a synthetic method for producing methoxy-NNO-azoxy compounds through the reaction of nitroso compounds with ammonium N-(methoxy)nitramide. The authors compare an electrochemical protocol employing a divided electrochemical cell with a purely chemical method and examine the substrate scope of both approaches. The structures of the obtained products are characterized using standard analytical techniques.
While the study addresses an interesting transformation, the overall yields are low in most cases, except for a limited number of substrates. As a report of a new synthetic methodology, the work remains insufficiently developed. In addition, the mechanistic discussion does not adequately explain why the reaction behavior varies significantly under the different conditions tested. Consequently, the manuscript does not yet satisfy the criteria required for publication of new synthetic methods in this journal. The authors should thoroughly address the following points.
The yields are consistently higher when either electron-donating or electron-withdrawing substituents are introduced at the meta position. The authors should provide a mechanistic rationale for this positional effect.
In several cases, the chemical method (Method B) affords higher yields than the electrochemical protocol. The authors should clarify what is occurring under these conditions. Is any additional information available beyond isolated yields (e.g., conversion, side-product profiles, reaction monitoring)?
The amounts of reagents used should be clearly indicated in the reaction schemes. Compounds 1a and 3 are used at 0.5 mmol and 2 mmol, respectively, in all entries, and this information should be explicitly included.
The authors should explain how and why the choice of electrode material influences the reaction outcome.
The presentation of Figures 1 and 2 is difficult to follow. The compounds corresponding to labels a–d should be directly indicated within the figures, and the observed potentials should also be reported.
Author Response
Referee: 2
Reviewer’s comment: The manuscript describes a synthetic method for producing methoxy-NNO-azoxy compounds through the reaction of nitroso compounds with ammonium N-(methoxy)nitramide. The authors compare an electrochemical protocol employing a divided electrochemical cell with a purely chemical method and examine the substrate scope of both approaches. The structures of the obtained products are characterized using standard analytical techniques.
While the study addresses an interesting transformation, the overall yields are low in most cases, except for a limited number of substrates. As a report of a new synthetic methodology, the work remains insufficiently developed. In addition, the mechanistic discussion does not adequately explain why the reaction behavior varies significantly under the different conditions tested. Consequently, the manuscript does not yet satisfy the criteria required for publication of new synthetic methods in this journal.
Answer: We thank the reviewer for his positive assessment of our manuscript and constructive criticism. Yields achieved by the proposed electrochemical method are indeed moderate for several substrates, however, as can be seen in Scheme 3, developed electrochemical method is currently the best available method for most nitroso compounds. Mechanistic discussion was improved according to comments of Reviewers 1 and 2.
Reviewer’s comment: The yields are consistently higher when either electron-donating or electron-withdrawing substituents are introduced at the meta position. The authors should provide a mechanistic rationale for this positional effect.
Answer: Thank you for this comment. Indeed, in the case of nitro-substituted nitrosobenzenes the highest yields were observed for the para- and meta-substituted products 2c and 2d. The lower yield for the ortho-substituted product 2b could be attributed to the steric hindrance around the nitroso group, which impedes its attack (including the effect of partial loss of conjugation with the phenyl ring due to steric repulsion between ortho-substituents). A similar result was obtained with nitrosobenzene 1j, containing a CF3-substitutient in the ortho-position of the benzene ring.
On the other hand, low yields were observed for products 2e and 2g, containing methoxy group in the ortho- and para-positions of the benzene ring. This can be explained by the positive mesomeric effect of the electron-donating methoxy group, which leads to a decrease in the oxidation potential of the corresponding nitroso compound and its early oxidation to the nitro derivative. When located in the meta-position of the benzene ring, the methoxy group is no longer conjugated with the nitroso group, and predominantly acts as an electron-withdrawing group via the inductive effect, which is consistent with increased yield of 2f compared to that of 2e and 2g. This explanation is in agreement with Hammett constants known for MeO- group: σm is about +0.115 (electron-withdrawing), σp is about -0.268 (electron-donating).
In case of the nucleophilic addition-elimination method higher yields of meta-substituted products could possibly be explained by the fact that mesomeric effects for this type of substitution are minimized, which leads to the reduction of the number of possible side processes. On the other hand, mesomeric effects of ortho- and para-substituents could play a significant role in the activation (in case of electron-withdrawing substituents) or deactivation (in case of electron-donating substituents) of the nitroso group, which could theoretically lead to the larger number of side processes or suppression of the primary reaction.
Corresponding text the discussion of Scheme 3 was corrected:
Old version: “The developed electrochemical methoxy-NNO-azoxylation protocol (Method A) was found to be applicable to both electron-deficient (1b–d) and electron-rich (1e–g) nitrosobenzenes, affording the corresponding products in moderate to good yields, while maintaining short reaction times (⁓ 1 h on a 0.5 mmol scale). The highest yields (53–92%) were obtained for compounds 2b–d prepared from nitrosobenzenes 1b–d containing an electron-withdrawing nitro group. Substitution with an electron-donating methoxy group furnished desired products 2e–g in generally moderate yields (33–87%). The higher yield of meta-substituted product 2f (87%) compared ortho- and para-substituted products 2e (51%) and 2g (33%) can be attributed to negative influence of electron-donating +M effect of methoxy group on the target product yield. Apparently, side processes of anodic oxidation of the nitroso group are favored by the conjugation between it and the electron-donating methoxy group.”
Revised version: “The developed electrochemical methoxy-NNO-azoxylation protocol (Method A) was found to be applicable to a range of electron-deficient (1b–d,j,k), electron-rich (1e–g), and halogen-substituted (2h,i) nitrosobenzenes, affording the corresponding products in moderate to good yields, while maintaining short reaction times (⁓ 1 h on a 0.5 mmol scale). The highest yields (51–92%) were obtained for compounds 2b–d,h,i,k prepared from nitrosobenzenes 1b–d,h,i,k containing either an electron-withdrawing nitro or trifluoromethyl group, or a halogen substituent. The decrease in yield in for ortho-substituted products (2b, 2h–2j) could be attributed to the steric hindrance around the nitroso group caused by the –NO2, –CF3 groups or halogen atoms. Substitution with an electron-donating methoxy group furnished the desired products 2e–g in generally moderate yields (33–87%). The higher yield of the meta-substituted product 2f (87%) compared to the ortho- and para-substituted products 2e (51%) and 2g (33%) can be explained by the positive mesomeric effect of the electron-donating methoxy group, which leads to a decrease in the oxidation potential of the corresponding nitroso compound and consequently its side processes of anodic oxidation. At the meta-position, the methoxy group does not conjugated with the nitroso group and thus predominantly acts as an electron-withdrawing group via the inductive effect, which is in agreement with its slightly positive Hammett constant σm about +0.1.”
Reviewer’s comment: In several cases, the chemical method (Method B) affords higher yields than the electrochemical protocol. The authors should clarify what is occurring under these conditions. Is any additional information available beyond isolated yields (e.g., conversion, side-product profiles, reaction monitoring)?
Answer: We thank the Reviewer for this essential note. It should be noted that chemical method and electrochemical protocol are based on the two fundamentally different mechanisms, which leads to them having different side processes. In the case of the chemical method, the main side process is probably the reduction of the nitroso group by the forming nitrite anion. In electrochemical protocol nitroso compound may be subjected to the wider amount of the possible side reactions. For instance, during electrochemical methoxy-NNO-azoxylation of 2-methyl-5-nitrosotetrazole 1s a substantial amount of the 5,5′-azoxy-bis-2,2′-methyltetrazole byproduct was formed. The latter data is now provided in the manuscript.
Added text: “In the case of the tetrazole derivative 1s, in addition to the target methoxy-NNO-azoxy product 2s, the known 5,5-azoxy-bis-2,2′-methyltetrazole (5) [72] was also isolated in 40% yield. Application of Method B enabled an increase in the yield of 2s from 28% to 51%, probably due to the mitigation of the competing process of formation of azoxytetrazole 5.”
Reviewer’s comment: The amounts of reagents used should be clearly indicated in the reaction schemes. Compounds 1a and 3 are used at 0.5 mmol and 2 mmol, respectively, in all entries, and this information should be explicitly included.
Answer: According to the Reviewer’s advice corresponding information was added in the reaction schemes.
Reviewer’s comment: The authors should explain how and why the choice of electrode material influences the reaction outcome.
Answer: We thank the Reviewer for this question. As can be seen from Table 3 (entries 15, 17–19), the cathode material has a negligible effect on the yield of 2a, because it is separated from the reaction mixture in a divided cell. Therefore, a stainless steel (SS) cathode was chosen as the most readily available material. On the other hand, the anode material also has a modest effect on the product yield in most cases. Using platinum, glassy carbon, and graphite anodes allowed us to obtain product 2a in 67 to 78% yield. The exception was the use of a nickel anode, which gave 2a in only 10% yield. We attribute this low yield to the oxidation of the nickel anode itself to Ni²⁺, a process commonly employed in sacrificial anodes. In response to the Reviewer's question, an additional experiment was conducted using carbon felt as the anode. With this material, 2a was also obtained in a high yield of 70%. So, except for Ni, many electrode materials are applicable in this process with comparable efficiency.
Reviewer’s comment: The presentation of Figures 1 and 2 is difficult to follow. The compounds corresponding to labels a–d should be directly indicated within the figures, and the observed potentials should also be reported.
Answer: According to the Reviewer’s advice, Figures 1 and 2 were corrected, corresponding structures and potentials were added.
Old versions:
(Figure 1)
(Figure 2)
Revised versions:
(Figure 1)
(Figure 2)

Round 2
Reviewer 2 Report
Comments and Suggestions for Authors
I appreciate the authors’ attempt to interpret the substituent effects using the Hammett equation. The discussion regarding the methoxy-substituted substrates is reasonable, and the difference between the meta and para cases can indeed be described in terms of σ constants. However, when examining other substituents, the correlation between σ values and the observed yields does not appear to be fully consistent.
For example, 3-NO₂ (σ = +0.71, 92%), 3-OMe (σ = +0.11, 87%), 4-NO₂ (σ = +0.78, 80%), and 3-CF₃ (σ = +0.43, 60%) do not align in a straightforward manner with the expected trends based solely on Hammett parameters. These cases suggest that additional factors may influence the reaction outcome.
In light of this, relying exclusively on a Hammett-type explanation for the methoxy-substituted series may not completely capture the overall behavior of the system. A more balanced or comprehensive discussion would strengthen the authors’ argument.
Author Response
Dear Reviewer,
thank you for careful reading of our manuscript and thoughtful feedback.
Reviewer's comment:
However, when examining other substituents, the correlation between σ values and the observed yields does not appear to be fully consistent.
For example, 3-NO₂ (σ = +0.71, 92%), 3-OMe (σ = +0.11, 87%), 4-NO₂ (σ = +0.78, 80%), and 3-CF₃ (σ = +0.43, 60%) do not align in a straightforward manner with the expected trends based solely on Hammett parameters. These cases suggest that additional factors may influence the reaction outcome.
In light of this, relying exclusively on a Hammett-type explanation for the methoxy-substituted series may not completely capture the overall behavior of the system. A more balanced or comprehensive discussion would strengthen the authors’ argument.
Answer: Thank you for pointing this out. We totally agree that the studied process is complicated, it involves multiple components, intermediates and processes, and thus its outcome cannot be predicted by just Hammett's σ values for a substituent in a specific nitrosoarene. The corresponding discussion was added to the manuscript:
Added text: "However, there is no strict correlation between electronic effects of substituents in meta- or para-positions and yields of the corresponding products 2. The yield is non-monotonously distributed in a series of substituents from the most electron-acceptor to the least electron-acceptor: p-NO₂ (σp = +0.78, 2d yield 80%), m-NO₂ (σm = +0.71, 2c yield 92%), m-CF₃ (σm = +0.43, 2k yield 60%), and m-OMe (σm = +0.11, 2f yield 87%). Apparently, there are several other factors determining the yield of target products 2 in electrochemical synthesis in addition to the major factor of nitrosoarenes stability to anodic oxidation, which may include: rate constants for the interception of N-(methoxy)nitramide-derived radical, tuning of a substituent electronic effect due to hydrogen bond formation, side processes associated with excessive electrophilicity."